# The burden of respiratory conditions in the emergency department of Muhimbili National Hospital in Tanzania in the first two years of the COVID-19 pandemic: A cross sectional descriptive study

Harrieth P. Ndumwa[1]*, Erick A. Mboya[1], Davis Elias Amani[1], Ramadhani Mashoka[2], Paulina Nicholaus[2], Rashan Haniffa[3,4,5], Abi Beane[3,4], Juma Mfinanga[2], Bruno Sunguya[1], Hendry R. Sawe[6], Tim Baker[6,7,8]

1 School of Public Health and Social Sciences, Muhimbili University of Health and Allied Sciences, Dar es Salaam, Tanzania, 2 Emergency Medicine Department, Muhimbili National Hospital, Dar es Salaam, Tanzania, 3 Mahidol Oxford Tropical Medicine Research Unit, Bangkok, Thailand, 4 Centre for Inflammation Research, University of Edinburgh, Edinburgh, United Kingdom, 5 University College London Hospitals, London, United Kingdom, 6 School of Medicine, Muhimbili University of Health and Allied Sciences, Dar es Salaam, Tanzania, 7 Department of Clinical Research, London School of Hygiene and Tropical Medicine, London, United Kingdom, 8 Department of Global Public Health, Karolinska Institutet, Solna, Sweden

* harrieth.peter@gmail.com

**Data Availability Statement:** Data supporting this publication are available within the manuscript, the

## Abstract

Globally, respiratory diseases cause 10 million deaths every year. With the COVID-19 pandemic, the burden of respiratory illness increased and led to significant morbidity and mortality in both high- and low-income countries. This study assessed the burden and trend of respiratory conditions among patients presenting to the emergency department of Muhimbili National Hospital in Tanzania and compared with national COVID-19 data to determine if this knowledge may be useful for the surveillance of disease outbreaks in settings of limited specific diagnostic testing. The study used routinely collected data from the electronic information system in the Emergency Medical Department (EMD) of Muhimbili National Hospital in Tanzania. All patients presenting to the EMD in a 2-year period, 2020 and 2021 with respiratory conditions were included. Descriptive statistics and graphical visualizations were used to describe the burden of respiratory conditions and the trends over time and to compare to national Tanzanian COVID-19 data during the same period. One in every four patients who presented to the EMD of the Muhimbili National Hospital had a respiratory condition– 1039 patients per month. Of the 24,942 patients, 52% were males, and the median age (IQR) was 34.7 (21.7, 53.7) years. The most common respiratory diagnoses were pneumonia (52%), upper respiratory tract infections (31%), asthma (4.8%) and suspected COVID-19 (2.5%). There were four peaks of respiratory conditions coinciding with the four waves in the national COVID-19 data. We conclude that the burden of respiratory conditions among patients presenting to the EMD of Muhimbili National Hospital is high. The trend shows four peaks of respiratory conditions in 2020–2021 seen to coincide with the four waves in the national COVID-19 data. Real-time hospital-based surveillance tools may be

raw data cannot be publicly shared due to ethical and legal reasons in Tanzania and as patients did not consent to have it shared in public servers. Upon request and following national policies in data sharing, data may be supplied to researchers requesting and fulfilling requirements as per the Tanzanian National Health Research Ethics guidelines. Contact information: National Health Research Ethics Review Committee Contact email: jikingura@nimr.or.tz Postal address: National Institute for Medical Research P.O. Box 9653 Dar es Salaam, Tanzania.

**Funding:** This study was supported by the African critical care registry network for pandemic surveillance, clinical management and research (MRC/UKRI MR/V030884/1)-RH, AB and TB; and by the Laerdal Foundation (2021-0097)-TB, HPN. HPN, EAM, RM and PN received salary support from the African critical care registry network for pandemic surveillance, clinical management and research funding for coordination and technical activities of the project. The funders had no role in study design, data collection and analysis, decision to publish, or preparation of the manuscript.

**Competing interests:** I have read the journal's policy and the authors of this manuscript have the following competing interests; TB declares a research grant and personal fees from the Wellcome Trust and personal fees from UNICEF, the World Bank, and USAID, all outside the submitted work. All other authors declare no competing interests.

useful for early detection of respiratory disease outbreaks and other public health emergencies in settings with limited diagnostic testing.

## Background

Respiratory conditions contribute the largest burden of disease and accounted for 10 million deaths globally in 2017 [1,2]. The five commonest respiratory conditions are asthma, chronic obstructive pulmonary diseases, acute respiratory infections, tuberculosis, and lung cancer [1,2]. In Africa, and in Tanzania, pneumonia, tuberculosis, and HIV-associated respiratory illnesses have been the most common respiratory conditions [3,4]. Between 2005 and 2015, about 13% of all deaths in Tanzania were due to respiratory illnesses [5,6]. Since 2020, the COVID-19 pandemic has caused a surge of respiratory disease and become a leading cause of respiratory morbidity and mortality [7]. Although the burden of COVID-19 has been seen to be less severe in Africa compared to the rest of the world, it nevertheless caused a severe strain to both health system and economic development, disrupting lives and social activities [8].

Despite the advancement of preventive measures such as childhood immunizations, improvements in diagnostics, and increased healthcare infrastructure in Tanzania, the trend of mortality from respiratory conditions has been increasing in the last decade [6]. Evidence of the burden and trend of respiratory conditions is not usually systematically collected in low- and middle- income countries (LMICs), including Tanzania, and surveillance tools may not be able to capture real-time data for disease monitoring. With the ongoing COVID-19 pandemic showing the vulnerability of health systems to public health emergencies and the challenges with good surveillance systems in low-resourced settings, hospital-based databases are regarded as a pragmatic tool that may help understand disease burdens and provide warnings of ongoing or impending threats, allowing the activation of response plans [9–11].

As in other LMICs, hospital-based surveillance systems in Tanzania are uncommon or are unable to give rapid warning signals to policy and decision makers. The primary aim of this study was to understand the burden of respiratory conditions among patients presenting to the EMD of Muhimbili National Hospital in Tanzania. Secondary aims were to describe the trends of respiratory conditions over a two-year period and to compare with national COVID-19 data to determine whether respiratory condition surveillance could be a useful tool for outbreaks and other public health emergencies in settings where the availability of diagnostic tests is limited.

## Methods

### Study design and setting

Routinely-collected clinical data of patients presenting to the Emergency Medical Department (EMD) of Muhimbili National Hospital (MNH) from 1$^{st}$ January 2020 to 31$^{st}$ December 2021 were used for this study. MNH is a 1500 bed facility that serves as the national referral hospital located in Dar es Salaam, the business capital of Tanzania with a population of 4 million [12]. Tanzania is a lower-middle-income country in East Africa with a population of 62 million [12] and a GDP per capita of US $ 1,076 [13]. Although the burden of COVID-19 may not have been as high in Tanzania as in other neighboring countries in the region, it nevertheless caused deaths, morbidities, and economic and social challenges [14].

The Muhimbili EMD has specialist emergency physicians and nurses trained in emergency and critical care, and is well resourced with equipment, medicines and supplies for treating

and stabilizing patients with acute and emergency conditions [15]. About 63,800 patients attend the Muhimbili EMD annually. The Muhimbili EMD uses an electronic information system to capture all patient information including registration records, investigations, prescriptions and for transferring patients to their respective admitting units, clinics or discharge. All EMD medical staff are trained and oriented to use the electronic system and fill in the patient information, diagnoses and other data on a routine basis. The data for this study were extracted from the EMD electronic information system and were anonymized prior to analysis.

### Study population, inclusion and exclusion criteria

All patients who presented with respiratory conditions to the Muhimbili EMD during the study period were included. A patient with a respiratory condition was defined as one of 1) a patient presenting with a respiratory complaint; 2) a patient given a respiratory diagnosis in the EMD; or 3) a patient with a deranged respiratory rate at initial triage. A respiratory complaint was defined as cough, difficulty in breathing, shortness of breath, chest pain, chest tightness or hemoptysis. A respiratory diagnosis was defined as pneumonia, upper respiratory tract infection (URTI), tuberculosis, lung cancer, chronic obstructive pulmonary disease (COPD), suspected COVID-19, asthma, bronchitis, bronchiolitis, tonsilitis, pharyngitis, sinusitis, laryngitis, rhinitis, emphysema, pyothorax, pneumothorax and influenza. Suspected COVID-19 was a diagnosis only used after March 20th 2020 when the first case was detected in Tanzania, and was a diagnosis made in the EMD based on clinical signs, laboratory and radiological results (Chest X-Ray) which pointed towards COVID-19, confirmatory tests could not be done. In the EMD, patients are often given more than one diagnosis, and in this study the first noted diagnosis was used. Deranged respiratory rate was defined as $< 8$ or $> 25$ per minute for patients aged 5 years and above. For young children, a deranged respiratory rate was defined as above 60 per minute for a child less than two months old, above 50 per minute for a child of 2–11 months and above 40 per minutes for a child of 1–5 years old [16]. While tachypnoea can be due to non-respiratory illness, respiratory causes are the most common underlying pathology in tachypnoea and we chose to include this criterion to ensure capturing of all respiratory illness as complaints are subjective and diagnoses in the EMD are preliminary. Patients who were dead on arrival at the EMD were not included in the study.

### Data extraction, management and analysis

Anonymized secondary data of all eligible patients were extracted from the electronic information system into STATA 17 (StataCorp), cleaned, and checked for completeness. Descriptive statistics for demographic characteristics were summarized using frequencies, percentages, median and IQR. The burden and trends of respiratory conditions in the two-year period were displayed with graphs. The common respiratory diagnoses were summarized in pie charts. The trends of the most common respiratory diagnoses; pneumonia, URTI, asthma and suspected COVID-19 were displayed in graphs. Additionally, a graph of confirmed COVID–19 in Tanzania mainland from March 2020 to December 2021 from the Tanzanian Ministry of Health COVID-19 Situation Report No 25, dated 4th March 2022 [14] was used for comparing the months of the peaks in our data of infectious diagnoses with national COVID-19 data.

### Ethical considerations

Ethical approval to conduct this study was obtained from the Muhimbili University of Health and Allied Sciences Ethical Review Board DA.282/298/06/C/767, MUHAS-REC-4-2020-217 and National Institute for Medical Research NIMR/HQ/R.8a/Vol.IX/3752. Permission to

extract patient data was obtained from Muhimbili National Hospital MNH/TRCU/IRB/Perm/ 2021/082. Since the study involved anonymized secondary data analysis, informed consent was not possible and its requirement was waived by the IRBs.

## Results

### The burden of respiratory conditions

In 2020–2021, 24,942 patients presented to the EMD of Muhimbili with a respiratory condition and were included in the study (Fig 1). This is one quarter (24.5%) of all patients presenting to the EMD. On average, 34 patients with respiratory conditions presented to the EMD per day, 1,039 patients per month.

Among the study participants, 12,999 (52.12%) were males. The age range was 2 days to 110 years with a median age (years) and IQR of 34.7 (21.7, 53.7). A quarter of all participants were aged less than 15 years. Three quarters of patients were from Dar es Salaam (Table 1).

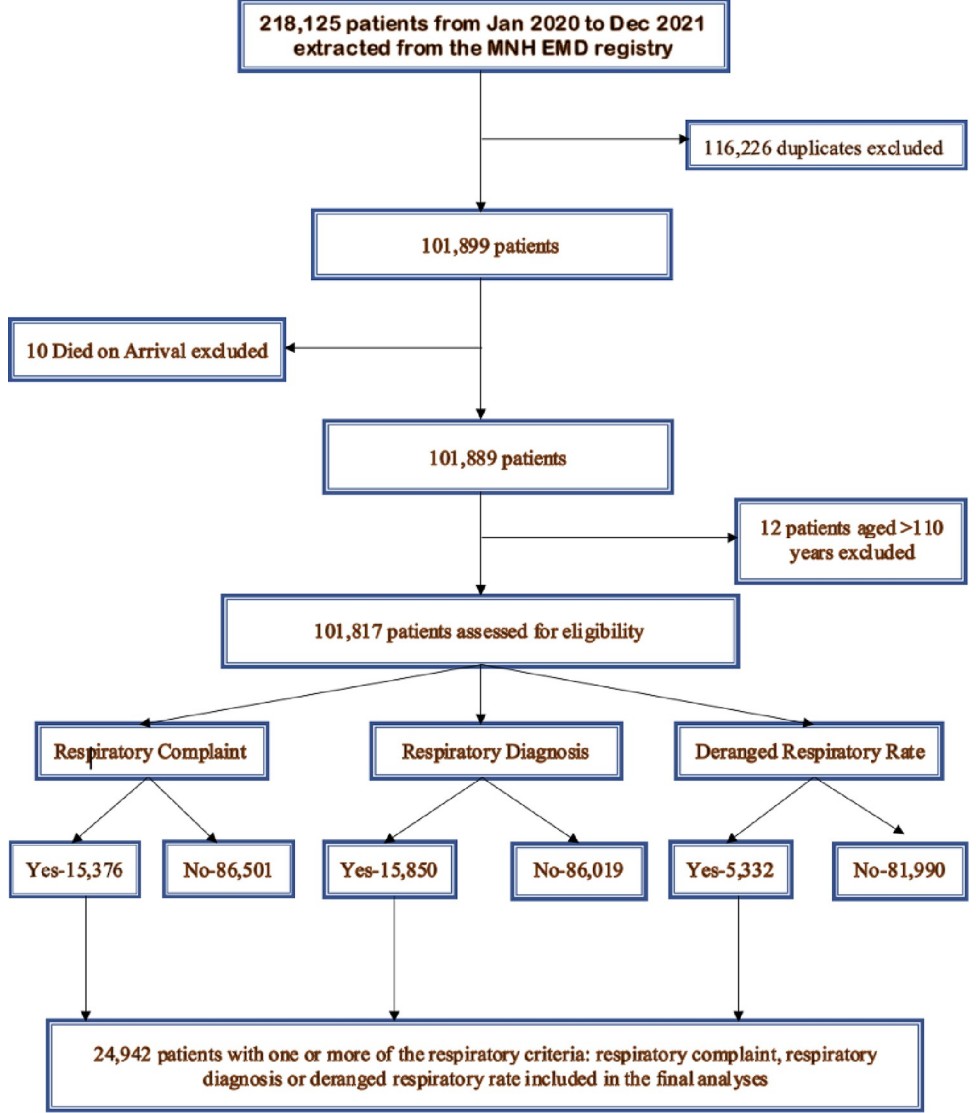

**Fig 1. Flow chart for inclusion in the study.**

**Table 1. Characteristics of patients with respiratory conditions at MNH EMD 2020–2021.**

| Characteristic | n | % |
|---|---|---|
| **Sex** | | |
| Male | 12999 | 52.12 |
| Female | 11939 | 47,87 |
| Unknown | 4 | 0.01 |
| **Age groups (in years)** | | |
| Median age, (IQR) | 34.7 (21.7, 53.7) | |
| 0 - <15 | 6369 | 25.54 |
| 15 - <30 | 4149 | 16.63 |
| 30 - <45 | 4878 | 19.56 |
| 45 - <60 | 4390 | 17.60 |
| 60 - <75 | 3650 | 14.63 |
| 75 - <90 | 1368 | 5.48 |
| 90+ | 138 | 0.55 |
| **Residency/Location** | | |
| Dar es Salaam | 18606 | 74.60 |
| Other parts of Tanzania Mainland | 5267 | 21.12 |
| Zanzibar | 190 | 0.76 |
| Outside Tanzania | 26 | 0.10 |
| Unknown | 853 | 3.21 |
| **Admitting Unit** | | |
| Pediatric | 2819 | 20.95 |
| Infectious and Respiratory Diseases Unit | 3438 | 25.54 |
| Medical Unit[1] | 3639 | 27.04 |
| Surgery[2] | 2862 | 21.27 |
| ICU | 700 | 5.20 |

[1]Includes: Cardiology, Dermatology, Endocrinology, Gastroenterology, Hematology, Nephrology, Neurology, Oncology and Psychiatry.

[2]Includes: Burns, ENT, Dental, Neurosurgery, Orthopedics, Gynecology, Obstetrics, Ophthalmology, Surgery and Urology.

## The weekly trend of respiratory conditions 2020–2021

In the study period, January 1st 2020 to December 31st 2021, the trend of respiratory conditions varied across weeks with some weeks having as few as 140 cases and other weeks having as many as 410 cases (Fig 2).

## The diagnoses and trends of common respiratory conditions at Muhimbili EMD

Of the patients in the study, 63% were given respiratory diagnoses (Fig 3A). The remaining patients presented with a respiratory complaint or deranged respiratory rate but were given non-respiratory diagnoses in the EMD. Half of the patients with a respiratory diagnosis had pneumonia, and upper respiratory tract infections were diagnosed in a third (31%) (Fig 3B). A few patients (10%) were given other respiratory diagnoses such as tuberculosis, COPD, lung tumor and emphysema.

## Trends of respiratory conditions

The trends of the most common respiratory diagnoses at EMD; pneumonia, Upper Respiratory Tract Infections (URTI), asthma and suspected COVID-19 are presented in Fig 4.

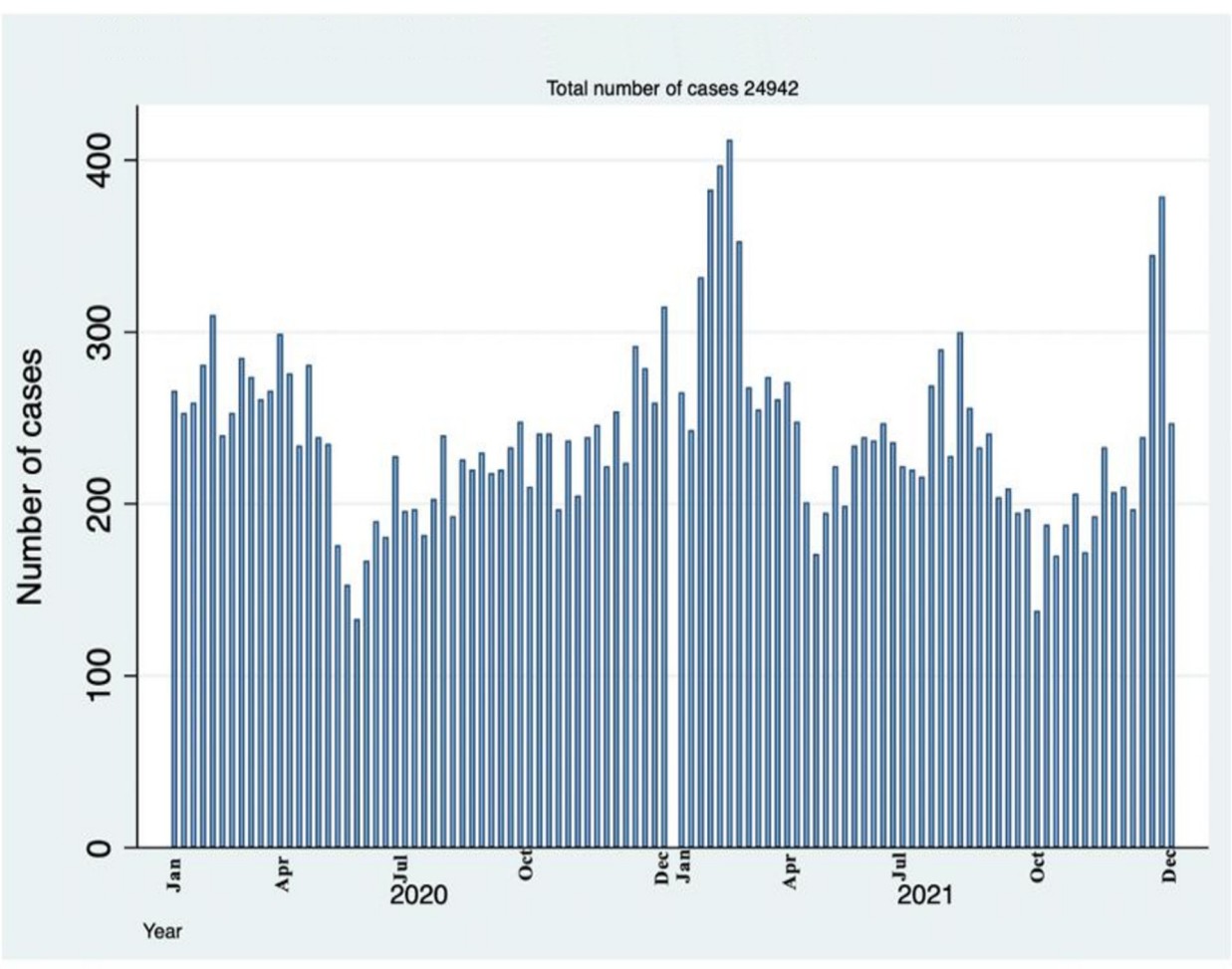

**Fig 2. Number of respiratory cases attended in a National Hospital in Tanzania by weeks.**

### Trends of respiratory conditions in comparison with the National COVID-19 data

The trends of infectious respiratory diagnoses (URTI, pneumonia and suspected COVID-19) are combined in Fig 5. Four peaks can be seen in April 2020, February 2021, August 2021 and December 2021. These peaks coincide with the months of the four waves of confirmed COVID-19 cases from the national COVID-19 data (Fig 6). The number of respiratory conditions appears to start increasing several weeks before the peaks in our data and in the national COVID-19 data (Figs 5 and 6).

### EMD outcomes

Of the study patients, nearly half 12,311 (49%) were admitted to hospital and 10,989 (44%) were discharged home. Four hundred (1.6%) of the patients died in the EMD (Fig 7).

### Discussion

This study has found that over 1000 patients per month–one in every four patients–present to an emergency department of a Tanzanian national hospital with a respiratory condition.

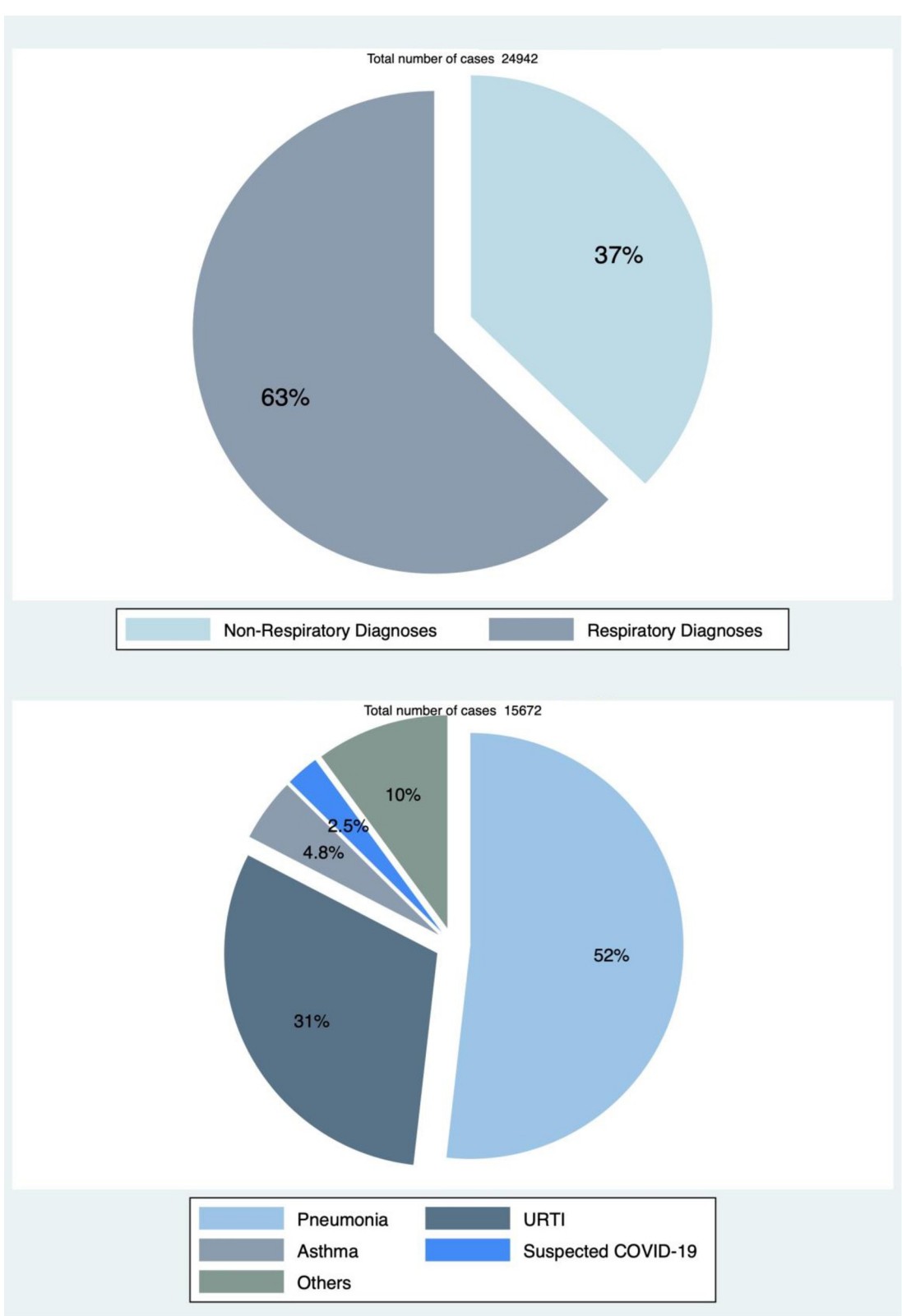

*Others include: Tuberculosis, COPD, Lung tumor and emphysema*

**Fig 3. Diagnoses of patients with respiratory conditions presenting to Muhimbili National Hospital.**

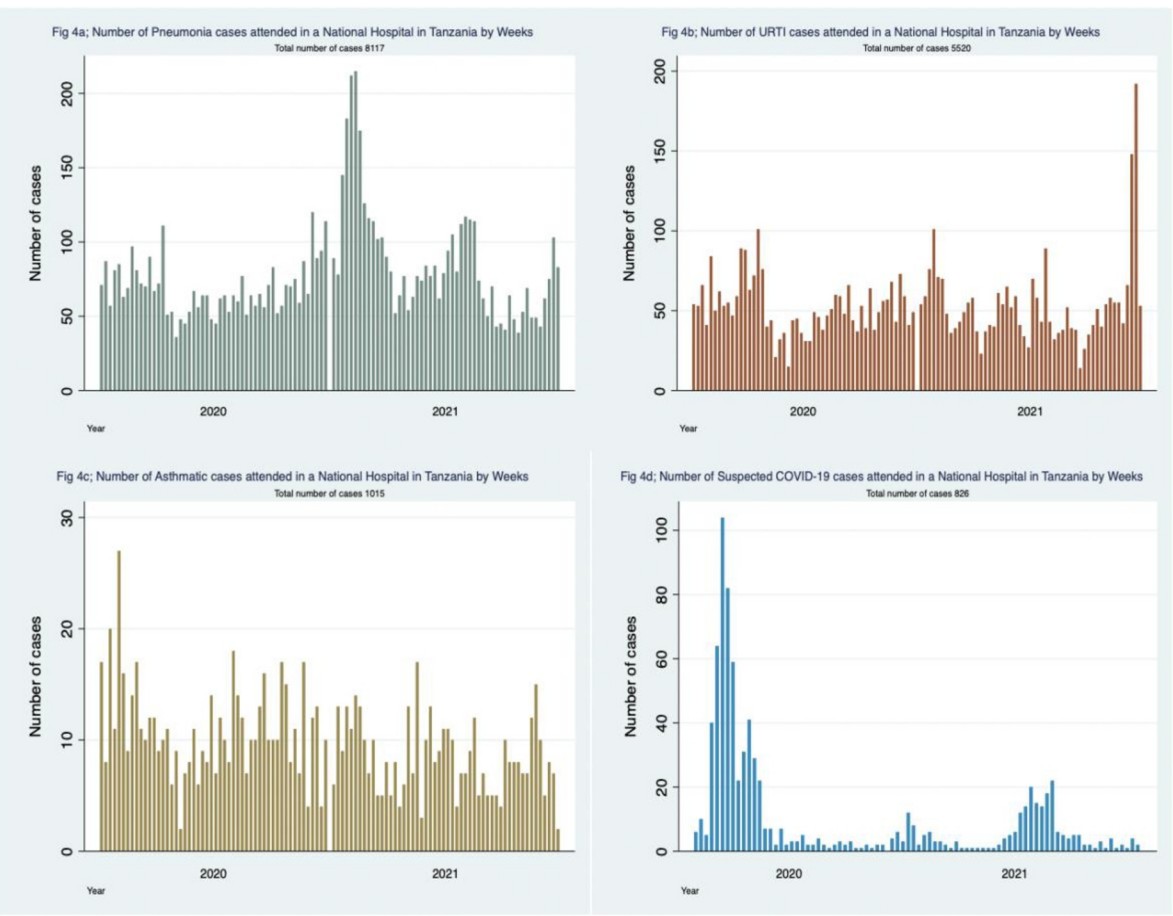

**Fig 4. Common respiratory diagnoses presented in a Tanzanian National Hospital.**

Respiratory conditions are a major contributor to the burden of disease in Tanzania. Similarly, respiratory conditions have been seen to lead to high proportions of healthcare contacts in the the United States [17] and visits to emergency departments in Canada [18]. Countries such as India have also reported disproportionately high burden particularly of chronic respiratory diseases [19]. In 2019, Tanzania was found to be one of the countries with the highest burden of acute lower respiratory tract infections with 1 in every 3 children under-5 years attending a health facility with a respiratory infection [5]. The WHO lists respiratory conditions as the second leading cause of death globally, following cardiovascular diseases [20]. Respiratory conditions cause a substantial burden on health systems, and even more so in the pandemic of the past two years.

The study was conducted in the context of the global COVID-19 pandemic. The trend of four peaks of patients presenting with respiratory conditions in 2020 to 2021 coincided with the four national COVID-19 waves suggests that the increase in respiratory conditions is likely to be due to COVID-19. We are not able to confirm this due to limited diagnostic testing and inability to cross-reference with laboratory data. As of 26 December 2021, over 278 million cases and just under 5.4 million deaths due to COVID-19 had been reported globally [21]. In the same period, 2020–2021, COVID-19 was the most common diagnosis in the emergency department in the UK, with fewer cases of other respiratory diagnoses [22].

The increase in respiratory conditions in our study starts about four weeks before the peaks seen in both our data and in the national COVID-19 data. This is unsurprising given the

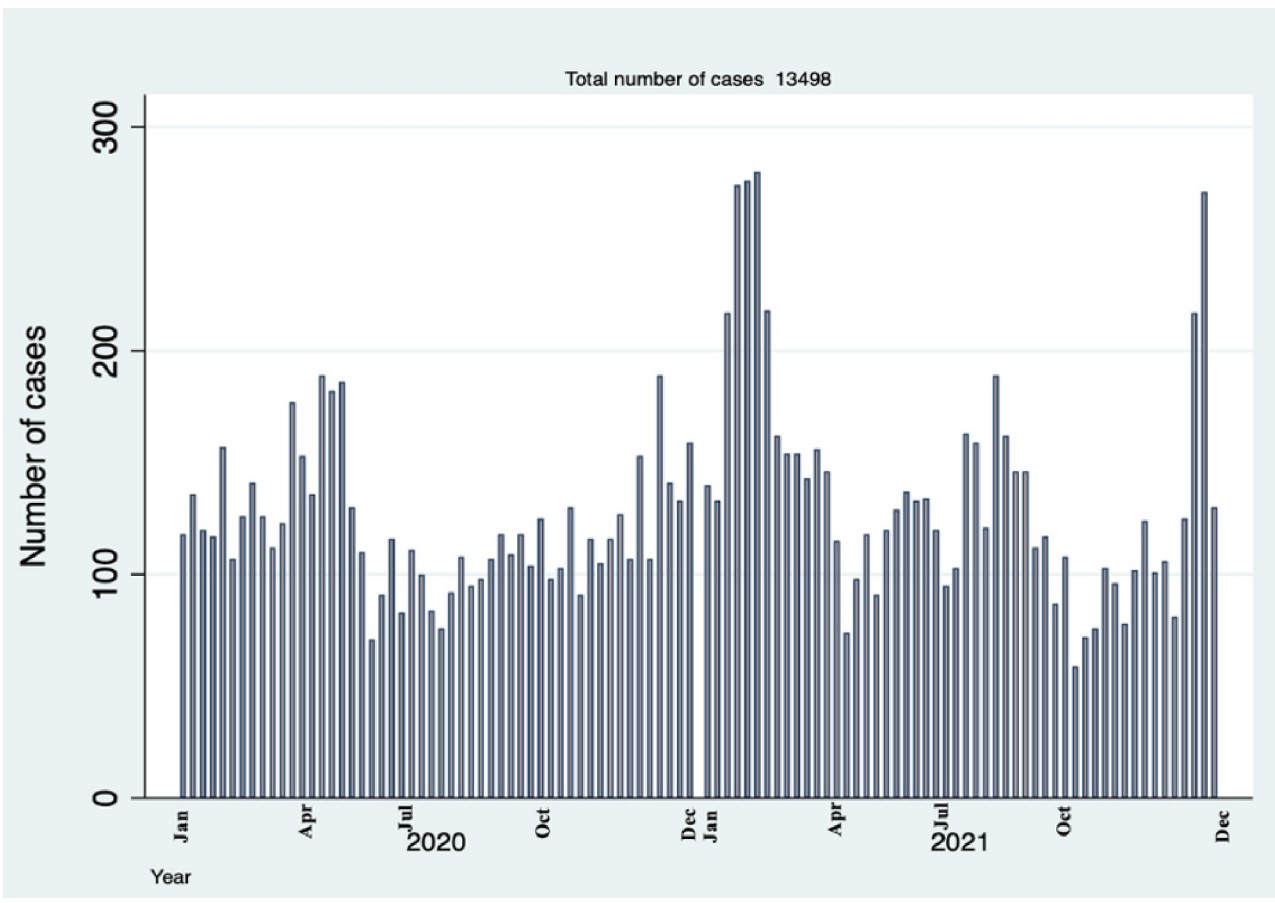

**Fig 5. Number of pneumonias, URT and suspected COVID-19 cases attended in a Tanzanian National Hospital by weeks.**

pattern of waves, and yet indicates an important public health usage of such respiratory surveillance. Clinical surveillance, even without specific diagnoses or confirmatory laboratory testing, can indicate when pandemic waves, or increases in disease due to any public health emergency, are starting and can inform the community and policy makers for the initiation of appropriate responses [9–11,23]. Interestingly, in Korea, patients with histories of URTIs and pneumonia were commonly reported 1–2 weeks before the COVID-19 pandemic was declared or COVID-19 tests were being routinely done [11,24]. Respiratory surveillance systems are being implemented in other areas such as the US and have proved to be beneficial [25,26].

Half of all patients with respiratory conditions were given a diagnosis of pneumonia which suggests that pneumonia may be the most common underlying etiology for respiratory conditions in this setting. These findings are similar with studies conducted elsewhere which found pneumonia to be a common etiology for respiratory illness [27,28]. Moreover, pneumonia followed by tuberculosis was the most common respiratory condition in Africa prior to the pandemic [3,4]. Similarly, a 10 years respiratory disease survey in Tanzania found pneumonia to be common and to account for more than half of deaths due to respiratory illnesses [6]. In other studies, the largest causes of respiratory illness were found to be asthma, chronic obstructive pulmonary diseases, acute respiratory infections, tuberculosis, and lung cancer [1,2]. Our study used the preliminary diagnosis given to patients in the EMD prior to definitive diagnostic testing and revised diagnoses, and pneumonia may be commonly used while waiting for the results of investigations.

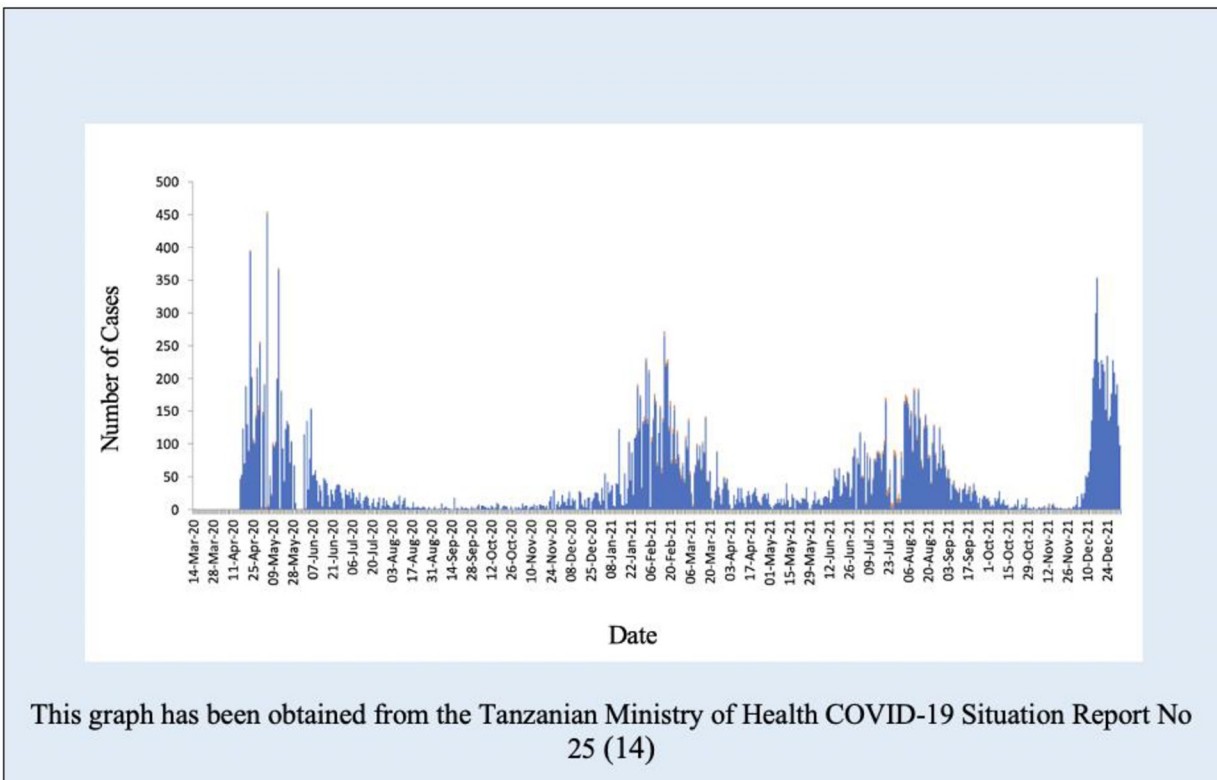

This graph has been obtained from the Tanzanian Ministry of Health COVID-19 Situation Report No 25 (14)

**Fig 6. COVID– 19 confirmed cases from March 2020 to December 2021 Tanzania Mainland.**

This is the first study conducted to assess the burden and trends of respiratory conditions in relation to the waves of COVID-19 pandemic in Tanzania. The study involved a large dataset of over 20,000 patients and was conducted at the largest hospital in Tanzania which receives patients from across the country. Additionally, the data used in this study has been prospectively collected as part of routine hospital system. Besides these strengths, some limitations are noted. The study involved retrospective analysis of prospectively collected data which was primarily for clinical purposes, hence limiting the analyses to the number of variables included in the registers. Besides being a national hospital and hence receiving patients from across Tanzania, this is a study of a single facility, as can be seen that majority of the patients were from Dar es Salaam, the location of the Muhimbili National Hospital and may therefore not reflect an overall situation in the country with regards to the pandemic outbreak. Moreover, the study was conducted at the emergency setting where the attention is on managing life threatening conditions rather than making definitive diagnoses, therefore, the respiratory diagnoses in this study are pragmatic rather than confirmatory and are subject to the treating physician's level of experience.

## Conclusion

The burden of respiratory conditions in the Muhimbili EMD in the first two years of the global COVID-19 pandemic was high, contributing to about a quarter of all presentations. The coinciding of the peaks of respiratory conditions with the four waves of confirmed COVID-19 cases in Tanzania suggests that they are likely to be due to COVID-19. Due to limited diagnostic testing, real-time patient surveillance in hospitals can be a useful tool for early detection of diseases outbreaks and other public health emergencies.

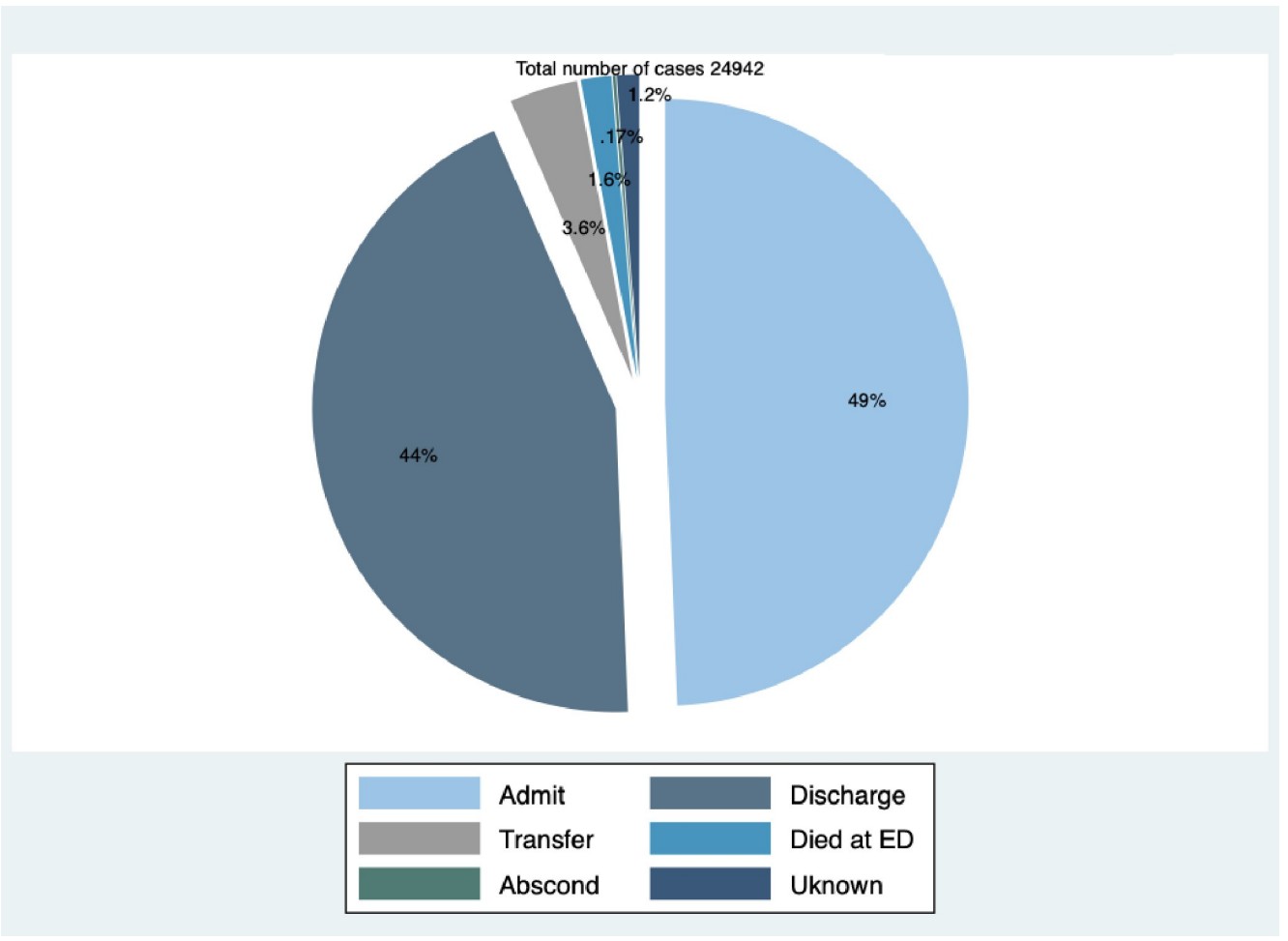

**Fig 7. Outcomes of respiratory cases attended in a National Hospital in Tanzania.**

## Acknowledgments

The authors wish to thank members of the Muhimbili National Hospital Emergency Department for their support and, the Directorate of Research and Publications of the Muhimbili University and Allied Sciences for providing conducive environment during proposal development and manuscript writing.

## Author Contributions

**Conceptualization:** Harrieth P. Ndumwa, Erick A. Mboya, Davis Elias Amani, Paulina Nicholaus, Rashan Haniffa, Abi Beane, Juma Mfinanga, Bruno Sunguya, Hendry R. Sawe, Tim Baker.

**Data curation:** Harrieth P. Ndumwa, Erick A. Mboya, Ramadhani Mashoka, Tim Baker.

**Formal analysis:** Harrieth P. Ndumwa, Erick A. Mboya, Tim Baker.

**Funding acquisition:** Harrieth P. Ndumwa, Rashan Haniffa, Abi Beane, Juma Mfinanga, Bruno Sunguya, Hendry R. Sawe, Tim Baker.

**Investigation:** Harrieth P. Ndumwa, Erick A. Mboya, Rashan Haniffa, Abi Beane, Juma Mfinanga, Bruno Sunguya, Hendry R. Sawe, Tim Baker.

**Methodology:** Harrieth P. Ndumwa, Erick A. Mboya, Juma Mfinanga, Bruno Sunguya, Hendry R. Sawe, Tim Baker.

**Project administration:** Harrieth P. Ndumwa, Ramadhani Mashoka, Paulina Nicholaus, Rashan Haniffa, Abi Beane, Juma Mfinanga, Bruno Sunguya, Hendry R. Sawe, Tim Baker.

**Resources:** Rashan Haniffa, Abi Beane, Juma Mfinanga, Bruno Sunguya, Hendry R. Sawe, Tim Baker.

**Supervision:** Harrieth P. Ndumwa, Juma Mfinanga, Bruno Sunguya, Hendry R. Sawe, Tim Baker.

**Validation:** Harrieth P. Ndumwa, Erick A. Mboya, Davis Elias Amani, Ramadhani Mashoka, Paulina Nicholaus, Rashan Haniffa, Abi Beane, Juma Mfinanga, Bruno Sunguya, Hendry R. Sawe, Tim Baker.

**Visualization:** Harrieth P. Ndumwa, Erick A. Mboya, Davis Elias Amani, Hendry R. Sawe, Tim Baker.

**Writing – original draft:** Harrieth P. Ndumwa, Tim Baker.

**Writing – review & editing:** Harrieth P. Ndumwa, Erick A. Mboya, Davis Elias Amani, Ramadhani Mashoka, Paulina Nicholaus, Rashan Haniffa, Abi Beane, Juma Mfinanga, Bruno Sunguya, Hendry R. Sawe, Tim Baker.

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
