## [Decision Letter · Decision Letter 0]

24 Aug 2022

PGPH-D-22-01035

The burden of respiratory conditions in the emergency department of Muhimbili National Hospital in Tanzania in the first two years of the COVID-19 pandemic

Dear Dr. Ndumwa,

Thank you for submitting your manuscript to PLOS Global Public Health. After careful consideration, we feel that it has merit but does not fully meet PLOS Global Public Health’s publication criteria as it currently stands. Therefore, we invite you to submit a revised version of the manuscript that addresses the points raised during the review process.

Please submit your revised manuscript by 10th September 2022. If you will need more time than this to complete your revisions, please reply to this message or contact the journal office at globalpubhealth@plos.org. Please include the following items when submitting your revised manuscript:

We look forward to receiving your revised manuscript.

Kind regards,

Reuben Kiggundu

Academic Editor

Journal Requirements:

1. We noticed that you have cited Figures 1-7 in the manuscript. However, there are no regular figures 1-7 but only supplementary figures. Please update the citations/labels to matched with the uploaded files.

2. In the online submission form, you indicated that "Data can be made available upon reasonable request from the corresponding author.". All PLOS journals now require all data underlying the findings described in their manuscript to be freely available to other researchers, either 1. In a public repository, 2. Within the manuscript itself, or 3. Uploaded as supplementary information.

3. Please update the 'Competing Interests' statement, including any COIs declared by your co-authors. If you have no competing interests to declare, please state "The authors have declared that no competing interests exist". Otherwise please declare all competing interests beginning with the statement "I have read the journal's policy and the authors of this manuscript have the following competing interests:"

4. Please amend your detailed Financial Disclosure statement. This is published with the article. It must therefore be completed in full sentences and contain the exact wording you wish to be published.

a. Please clarify all sources of funding (financial or material support) for your study. List the grants (with grant number) or organizations (with url) that supported your study, including funding received from your institution. 

b. State the initials, alongside each funding source, of each author to receive each grant.

c. State what role the funders took in the study. If the funders had no role in your study, please state: “The funders had no role in study design, data collection and analysis, decision to publish, or preparation of the manuscript.”

d. If any authors received a salary from any of your funders, please state which authors and which funders.

Reviewers' comments:

Reviewer's Responses to Questions

**Comments to the Author**

1. Does this manuscript meet PLOS Global Public Health’s publication criteria? Is the manuscript technically sound, and do the data support the conclusions? The manuscript must describe methodologically and ethically rigorous research with conclusions that are appropriately drawn based on the data presented.

Reviewer #1: Partly

2. Has the statistical analysis been performed appropriately and rigorously?

Reviewer #1: N/A

3. Have the authors made all data underlying the findings in their manuscript fully available (please refer to the Data Availability Statement at the start of the manuscript PDF file)?

Reviewer #1: Yes

4. Is the manuscript presented in an intelligible fashion and written in standard English?

Reviewer #1: Yes

5. Review Comments to the Author

Reviewer #1: i want to commend the researchers in their efforts to contribute to scientific knowledge.

However, I feel that overall the paper is not contributing new information. The conclusion of the paper simply stated is that respiratory illness in their center peaked during the first two years of the respiratory pandemic, which is what was seen around the world.

From my review, I believe the work is original. I can’t comment on whether it has been published elsewhere. this is an observational study, hence, no experiments were carried out.

I do feel that the conclusion need to be revisited, as it states that 52% of the total respiratory cases seen were due to pneumonia while 2.5% were due to COVID-19 [which is also a form of pneumonia ]. the paper did not state how it differentiated COVID-19 pneumonia from other causes of pneumonia. the authors wrote that testing capacity was limited but did not mention the use of case definitions which can be used to identify cases in an epidemic/pandemic.

I recommend the research title include the design according to the STROBE checklist for manuscript writing

I also recommend that the limitations of the study be exhaustively stated.

6. PLOS authors have the option to publish the peer review history of their article (what does this mean?). If published, this will include your full peer review and any attached files.

**Do you want your identity to be public for this peer review?** For information about this choice, including consent withdrawal, please see our Privacy Policy.

Reviewer #1: No

---

## [Editor Report · Decision Letter 1]

7 Nov 2022

The burden of respiratory conditions in the emergency department of Muhimbili National Hospital in Tanzania in the first two years of the COVID-19 pandemic; A cross sectional descriptive study.

PGPH-D-22-01035R1

Dear Dr Ndumwa,

We are pleased to inform you that your manuscript 'The burden of respiratory conditions in the emergency department of Muhimbili National Hospital in Tanzania in the first two years of the COVID-19 pandemic; A cross sectional descriptive study.' has been provisionally accepted for publication in PLOS Global Public Health.

Best regards,

Reuben Kiggundu

Academic Editor